# Refractive Characteristics and Related Factors of Amblyopia after Lamellar Keratoscleroplasty in Children with Limbal Dermoids

**DOI:** 10.3390/jcm11144176

**Published:** 2022-07-19

**Authors:** Surong Luo, Jianjiang Xu, Tingting Shao, Xiaomei Qu

**Affiliations:** 1Department of Ophthalmology, Shaoxing People’s Hospital (Shaoxing Hospital of Zhejiang University), Shaoxing 312000, China; luosurong520@163.com; 2Department of Ophthalmology and Vision Science, Eye & ENT Hospital, Fudan University, Shanghai 200031, China; jianjiangxu@126.com (J.X.); shaotingting@fudan.edu.cn (T.S.); 3NHC Key Laboratory of Myopia, Fudan University, Shanghai 200031, China; 4Laboratory of Myopia, Chinese Academy of Medical Sciences, Shanghai 200031, China

**Keywords:** limbal dermoids, lamellar keratoscleroplasty, corneal topography, refractive error, amblyopia

## Abstract

We examined the refractive characteristics and related factors of amblyopia in pediatric patients with limbal dermoids undergoing lamellar keratoscleroplasty. Forty-one children (mean age: 56.15 ± 22.47 months) were enrolled. Cycloplegic refraction, corneal topography, and anterior segment photography were performed. The corneal topographic and distribution characteristics of the refractive state were summarized, and the relationship between limbal dermoid invasion size and the refractive state was analyzed. The relationship between invasion size and amblyopia severity and the effect of clinical intervention at different times on amblyopia treatment were also analyzed. The spherical power distribution was −1.0–+10.75 D (average: +4.79 ± 3.09 D). The cylinder power was −1.25–−8.75 D (average: −4.19 ± 1.93 D). The axial range of astigmatism was 10–180° (average: 103.54 ± 58.16°). Equivalent spherical refraction was −3.88–+7.76 D (average: +2.70 ± 3.08 D). Twenty-five, fifteen, and one case had limboid dermoid invasion of the central circular zone (CCZ), paracentral annular zone (PCZ), and corneal limbus within 1 mm, respectively. Corneal topography of 39 patients showed flat, steep, and mean curvatures of 38.48 ± 2.12 D, 43.29 ± 1.97 D, and 40.70 ± 1.48 D, respectively. The mean astigmatism was 4.80 ± 2.93 D in the 3-mm optical region. Astigmatism was higher in CCZ than in PCZ invasion (*p* < 0.05). Postoperative visual acuity was positively correlated with patients’ age and amblyopia treatment duration (r = 0.392, *p* = 0.048; r = 0.488, *p* = 0.011), and was negatively correlated with astigmatism (r = −0.646, *p* < 0.001). High hyperopia and astigmatism are the dominant refractive errors in patients with limbal dermoids undergoing lamellar keratoscleroplasty.

## 1. Introduction

A corneal limbal dermoid is a congenital choristoma that exists at birth [1] and it can gradually increase with age [2]. Corneal limbal dermoid also has a relatively large impact on vision [3]. Surgery for limbal dermoid has been very successful by improving appearance and rarely causing rejection [4,5,6]. However, how corneal morphology, refraction, and vision recovery change after lamellar keratoscleroplasty remains unknown. Therefore, in this study, we summarized the refractive state of patients after surgery, and analyzed their corneal morphology and refractive state characteristics. Our study aims included the following: (1) to study the relationship between the size of limbal dermoid invasion, corneal morphology, and refractive state; (2) to study the relationship between these variables and amblyopia formation; and (3) to summarize the clinical effects of different clinical intervention times and durations after amblyopia formation.

## 2. Materials and Methods

### 2.1. Patients

A cross-sectional study was performed for 41 patients (41 eyes; sex, 21 male and 21 female patients; mean age, 56.15 ± 22.47 (range: 22.7–117.1) months) who were pathologically diagnosed with limbal dermoids and who underwent lamellar keratoscleroplasty at the Eye, Ear, Nose, and Throat Hospital of Fudan University in Shanghai. These patients came to the outpatient clinic for follow-up from September 2020 to April 2021. All of the enrolled patients had a single eye corneal limbal dermoid. We examined the eye with the corneal limbal dermoid. Basic information is shown in Table 1. This study complied with the Declaration of Helsinki, and the parents of all patients were informed of the details of the study and provided their informed consent. The study was approved by the hospital’s Ethics Committee on 8 June 2015.

### 2.2. Instrument and Methods

#### 2.2.1. Vision

A standard logarithmic visual acuity chart was used to check corrected visual acuity in the naked eyes of those who could cooperate with the examination. Routine anterior segment examination and fundoscopic examination were performed on all patients to exclude other organic ocular lesions. Cycloplegic refraction was used to assess refractive errors. Cycloplegia was achieved using four drops of compound tropicamide eye drops (mixture of 0.5% tropicamide and 0.5% phenylephrine eye drops; Santen Pharmaceutical Co., Ltd., Osaka, Japan), administered approximately 5 min apart. Cycloplegic autorefraction was measured at 30 min after the last drop using a desktop autorefractor (KR-8800; Topcon Corporation, Tokyo, Japan). Three readings with 0.25 Diopter (D) or less apart in both the spherical and cylindrical components were averaged. Then, cycloplegic retinoscopy was performed by an experienced optometrist. According to the consensus of experts on the Prevention and Treatment of Amblyopia in Chinese Children (2021) [7], to exclude ocular organic disease, the normal lower limit of vision is 0.5 for children aged 3–5 years and 0.7 for children aged ≥6 years. If the corrected visual acuity is lower than this limit or if the difference of visual acuity is two rows or more, amblyopia is considered.

#### 2.2.2. Pentacam Anterior Segment Analysis System

All patients who could cooperate were examined by Pentacam corneal topography (Oculus, Wetzlar, Germany). Two patients were too young to complete the corneal topography examination, and therefore, we completed the corneal topography examination in 39 patients.

The Pentacam is an automatic non-contact device that utilizes a rotating Scheimpflug camera and monochromatic slit illumination to provide video-based automated anterior segment measurements. The camera rotates around the eye while taking images, and with the help of an internal software, it generates a three-dimensional model of the anterior segment. The anterior segment model consists of high-resolution imaging that allows for assessment of the entire surface structure of the cornea, as well as the measurement of the anterior chamber depth. Pentacam performed the measurement automatically when correct alignment with the corneal apex and focus was achieved. The noninvasive measurement process of the Pentacam requires 2 s, and collects 25–50 single captures that consist of 13,800 true elevation points while rotating around the optical axis of the eye. This results in a Pentacam-built three-dimensional model of the entire anterior eye segment.

Records of patients examined with the Scheimpflug principle-based Pentacam corneal tomographer were reviewed for topographic parameters. Parameters studied included keratometry readings, topographic astigmatism, corneal eccentricity in the central 6 mm, average radius of curvature between the 6-mm and 9-mm zone center, and minimum sagittal curvature for anterior and posterior cornea surfaces.

Multiple measurements were made, and the images with the best quality were selected for data collection. Collection parameters were as follows: plane center curvature, K1; steep center curvature, K2; mean corneal curvature, Km; and mean astigmatism, Ast in the 3-mm optical region; maximum corneal curvature, CCmax; minimum corneal curvature, CCmin; maximum corneal thickness, CTmax; and maximum anterior surface height, ASHmax. Within the limbal dermoid invasion range, the parameters of corneal curvature, corneal thickness, and anterior surface height at the corresponding position on the diagonal were collected. Based on the image report, the corneal area invaded by limbal dermoid and the number of hours involved were recorded, alongside whether the cyst invaded the central circular zone (CCZ: referring to the 5-mm area of the central cornea) or the paracentral annular zone (PCZ: the 5–8-mm annular zone outside the CCZ) [8]. Invasion of the central and paracentral cornea was evaluated by Pentacam corneal topography. The limbal dermoids invading the corneal range and the number of hours are presented in Table 2.

The patients were divided into two groups according to the extent of limbal dermoid invasion: 25 cases (25 eyes) with CCZ invasion were included in the CCZ group, whereas 15 cases of PCZ invasion (15 eyes) and a case of corneal limbal invasion within 1 mm (one eye) were included in the PCZ group. The refractive state of the two groups was compared. Among them, two patients could not cooperate with the corneal topographic examination due to age-related factors. For these patients, we used a slit-lamp microscope to observe and record the corneal invasion range of the limbal dermoids; then, these patients were enrolled into the corresponding CCZ or PCZ group.

### 2.3. Statistical Analysis

Data were analyzed using SPSS statistical software version 25 (IBM Corp, Armonk, NY, USA). The mean and standard deviation, range, and percentage were used to express the data. All data in this study were tested by the Shapiro–Wilk test to determine whether they were normally distributed. The corneal topographic parameters (CCmax, CCmin, CTmax, and ASHmax) that followed the normal distribution were compared by a paired sample *t* test. An independent sample t-test was used to compare the data for statistical differences according to the invasion range and treatment duration of limbal dermoids (Levene’s variance test was performed to judge the homogeneity of variance; if the significance was 0.05, the variance was uniform (see the result of “assumed isovariance” in the first line); if the significance was <0.05, the variance was uneven (see the result of “equivariance is not assumed” in the second line)). The correlation between amblyopia and factors that may affect visual recovery was analyzed by Pearson’s correlation analysis. A *p*-value < 0.05 was considered statistically significant, with all reported *p*-values as two sided.

## 3. Results

### 3.1. Corneal Topography of 39 Patients with Limbal Dermoids

K1, K2, Km, and Ast are presented in Table 3. The mean value of corneal astigmatism was 4.80 ± 2.93 D, the span of astigmatism was 10.8 D, and the variance value was 8.59, suggesting that patients with limbal dermoids after lamellar keratoscleroplasty had high average astigmatism. The distribution span and discrete value of astigmatism were also large, suggesting a great influence on the central morphology of the cornea.

### 3.2. Anterior Corneal Surface Parameters

A limbal dermoid presents as a hemispherical or fusiform bulging mass at the margin of the corneal limbus. The extent of corneal invasion directly affects the morphology of the cornea, resulting in changes in corneal curvature, thickness, and height. Abnormal corneal morphology leads to abnormal diopters; lamellar keratoscleroplasty does not completely change this morphology. We recorded CCmax, CCmin, CTmax, and ASHmax within the range of limbal dermoids in patients after lamellar keratoscleroplasty. The parameters are compared with those obtained at the corresponding positions on the diagonal (Table 4).

The statistical results show the following: (1) For patients with limbal dermoids after lamellar keratoscleroplasty, the maximum curvature value of the invaded area was significantly different from its diagonal curvature. It may be because of the relationship between donor and recipient at the graft edge that the corneal morphology changes, and because of the tightness of the corneal sutures, resulting in the inconsistency of maximum corneal curvature in the range of limbal dermoids; (2) There was no statistical difference between the minimum curvature value of the limbal dermoid invasion area and the diagonal curvature. One possible reason is that the corneal curvature at the edge of the graft was significantly affected after the orthotopic sutured lamellar corneal graft, whereas the corneal curvature at the center of the graft was not, and the graft itself was smooth; (3) The corneal thickness and maximum anterior surface height within the invaded area were significantly different from those at the corresponding position on the diagonal. This may be related to corneal edema caused by immersion during the preservation of the in vitro corneal graft and excessive thickness of the intraoperative grafts.

### 3.3. Comparison of Refractive State between the CCZ Group and PCZ Group

Patients were grouped according to the extent of limbal dermoid invasion. Those in the CCZ group experienced invasion in the 5-mm area of the central cornea, whereas those in the PCZ group experienced invasion in the 5–8-mm annular zone outside the CCZ. The diopter distribution of the two groups is presented in Table 5.

The results show that there was a significant statistical difference between the CCZ and PCZ groups in the number of hours involved. Moreover, astigmatism was higher in the CCZ than in the PCZ group (*p* < 0.05). Finally, there were no significant differences in diopter and equivalent spherical lens refraction between the two groups (all *p* > 0.05).

### 3.4. Duration of Amblyopia Treatment

If the patients had abnormal vision or an abnormal refractive state after surgery, intervention measures, such as wearing glasses or covering their eyes, were introduced. These patients were split into two groups according to the duration of the intervention (treatment duration >12 vs. <12 months). The results showed that there was no significant difference in diopters between the two groups (Table 6).

### 3.5. Factors Related to Visual Recovery

Fourteen patients could not cooperate with the visual acuity examination, and 27 patients could. Based on the consensus of experts on the Prevention and Treatment of Amblyopia in Chinese Children (2021) [7], 18 cases of amblyopia were diagnosed among the 27 cases who could cooperate, and the remaining nine cases met the vision standard of this age group. We used Pearson’s correlation analysis to analyze the factors that may influence the formation of amblyopia in the 27 patients who could complete the examination. These factors included age, surgical age, the extent of limbal dermoid invasion (CCZ or PCZ), the number of hours involved, the presence of amblyopia treatment intervention, time between amblyopia intervention and surgery, the duration of amblyopia treatment intervention, spherical lens refraction, astigmatism, axis of astigmatism, and equivalent spherical diameter. The results showed that visual acuity recovery was correlated with patients’ age (r = 0.392, *p* = 0.048), treatment duration (r = 0.488, *p* = 0.011), and astigmatism (r = −0.646, *p* < 0.001); there was no correlation with other factors (Table 7).

## 4. Discussion

The infant period is a critical period for human visual development. From 9 months to 2 years after birth, infants are most susceptible to abnormal vision; the sensitivity and plasticity of visual development gradually decreases from the age of 2 years to the age of 8 years [9,10]. Owing to infants’ limited cognition and language expression, their early abnormal vision is often ignored. However, for children with limbal dermoids, the abnormal appearance of the eye surface can remind parents to pay attention and take their children to the hospital early. Factors such as the size of the corneal area invaded by the limbal dermoid; whether there are ocular, ear, or systemic complications; and parents’ awareness and attention to the disease may influence whether lamellar keratoscleroplasty is used in infants with limbal dermoids that do not invade the whole cornea. Early postoperative visual examination, detection of abnormal visual development, and treatment of abnormal visual acuity should be initiated by the patient’s parents. Abnormal visual development in infants and young children needs to be recognized early and corrected in time; a younger patient age increases the sensitivity to amblyopia treatment.

Astigmatism, refractive amblyopia, and low vision caused by limbal dermoids are considered as indications for surgical treatment and postoperative problems [5]. Moreover, surgery can improve the patient’s appearance and achieve their parents’ desired cosmetic effect [5,11]. Yamashita et al. mentioned that lamellar keratoscleroplasty with preserved corneas was performed for patients with limbal dermoids, with good postoperative cosmetic results [11]. However, limbal dermoids lead to changes in corneal morphology and refraction, and the improvement of the corneal astigmatism and visual acuity of patients after lamellar keratoscleroplasty cannot be guaranteed. The article also mentioned that preoperative treatment of amblyopia can improve postoperative visual acuity, and long-term observation and follow-up examinations are required for preoperative and postoperative amblyopia treatment.

During the postoperative follow-up period, the patients were examined by Pentacam corneal topography and anterior segment photography. Their visual acuity, spherical lens refraction, and astigmatism were recorded. The corneal topographic characteristics and distribution characteristics of the refractive state were summarized, and the relationship between the extent of limbal dermoid invasion, corneal morphology, and refractive state was analyzed. Furthermore, the relationship between the size of the limbal dermoid invasion and amblyopia formation, as well as the effect of clinical intervention at different times after amblyopia formation were also examined. We found that most patients experienced hyperopia, which may be partly attributed to the normal visual development of young children. The clinical presentation of a limbal dermoid is a hemispherical or fusiform bulging mass at the margin of the corneal limbus. The extent of corneal invasion directly affects the morphology of the cornea, and abnormal corneal morphology leads to abnormal diopters. Lamellar keratoscleroplasty does not completely correct the abnormal corneal morphology and refraction. Therefore, high astigmatism is mostly caused by corneal morphological changes because of the presence of limbal dermoids in patients. Hussein et al. [12] concluded that initial low vision, age >6 years, obvious astigmatism, and poor treatment compliance are the factors that can lead to the failure of anisometropic amblyopia treatment. However, in a paper published by the ophthalmic surface team of the Fudan University Eye, Ear, Nose, and Throat Hospital [13], it was found that residual and surgically-induced equivalent spherical diameter were both reduced in long-term amblyopia treatment, but the correction effect on postoperative astigmatism was disappointing. As astigmatism and hyperopia may cause anisometropia or anisometropic amblyopia, we should use standard corrective glasses and occlusion therapy as soon as possible to correct these conditions [14].

In an article by Kim et al. [15], geometric parameters were designed for preoperative evaluation of limbal dermoid lesions to predict surgical results. The results showed that the ratio of the wide axial dermoid length to the corneal diameter, that is, the scope of limbal dermoid invasion, is the most important factor affecting postoperative vision, amblyopia development, and postoperative scar formation. Thus, early surgical intervention may be a well-intentioned option for patients with progressive limbal dermoid size. In another retrospective study [3], the authors established a new visual grading system based on the corneal area, surface shape, and conjunctiva area. This system has a good predictive value for pathology and prognosis, suggesting that the pathological type of low-level dermoid lesions is relatively mild, and postoperative vision is better. Corneal limbal dermoids can interfere with vision by inducing irregular astigmatism or invading the pupillary area, thereby leading to refractive or deprivational amblyopia [16]. Therefore, preoperative and postoperative intervention for visual recovery of limbal dermoids is particularly important. According to our data analysis, the postoperative visual acuity recovery of patients with limbal dermoids was positively correlated with their age (r = 0.392) and amblyopia treatment duration (r = 0.488). Nevertheless, it was negatively correlated with astigmatism (r = −0.646). Considering that the aging of younger patients is accompanied by improved visual acuity, a longer amblyopia intervention (mainly including wearing correction glasses and eye covering) may result in a more positive visual acuity recovery effect. It is understandable that a higher astigmatism may result in worse visual recovery. In terms of recovery of amblyopia and improvement of visual acuity, early surgical intervention did not show positive results if limbal dermoids were stable in size. Therefore, patients with limbal dermoids without aesthetic considerations may not prioritize surgery, but aggressive amblyopia treatment should be initiated prior to surgery. In conclusion, for patients with ametropic amblyopia caused by limbal dermoids, the treatment of amblyopia needs early intervention before surgery. The timing of surgery should consider the size and appearance of the cyst, as well as parental awareness and other factors.

As surgery cannot change the high refractive abnormality, especially the astigmatism, we should seek ways to reduce the astigmatism of patients, including adjusting the tightness of sutures and removal of intermittent stitches, as well as postoperative optical correction methods (i.e., wearing RGP lenses) and active amblyopia treatment methods (i.e., perceptual learning) [17].

### Limitations

Our study has some limitations. First, it was a cross-sectional study, and the relatively small sample size may have limited our ability to reach definitive conclusions. Second, the patients’ operation age was relatively young, and some patients might have been lost to follow-up. Therefore, there were no complete refractive data available both before and after surgery. Here, only the results of one outpatient examination of patients with limbal dermoids were selected, and the comparison of preoperative and postoperative diopters was not conducted. Third, long-term observations are required to expand the knowledge concerning the vision results of patients with limbal dermoids.

## 5. Conclusions

Patients with limbal dermoids still have a high degree of astigmatism after their operation, which is the main cause of postoperative amblyopia. Surgery cannot change the degree of astigmatism, so it may not be the option of choice; instead, active preoperative and postoperative amblyopia treatment are the key.

## Figures and Tables

**Table 1 jcm-11-04176-t001:** Baseline characteristics of 41 children with limbal dermoids.

Variable	Mean ± SD	Range/%
Sex		
Male, *n*	21	51.22%
Female, *n*	20	48.78%
Age (months), mean	56.15 ± 22.47	22.7–117.1
Surgical age (months), mean	25.96 ± 11.46	4.2–54.1
Amblyopia		
Yes	18	43.90%
No	9	21.95%
Unable to check vision	14	34.15%
Amblyopia treatment		
Yes	18	43.90%
No	23	56.10%
Duration between amblyopia intervention and operation time (months), mean	19.40 ± 14.96	1.4–63.0
Amblyopia treatment duration (months), mean	10.79 ± 20.41	0.0–82.2
Spherical (D), mean	+4.79 ± 3.09	−1.00–+10.75
Astigmatism (D), mean	−4.19 ± 1.93	−1.25–−8.75
Axis of astigmatism (degrees), mean	103.54 ± 58.16	10–180
SE (D), mean	+2.70 ± 3.08	−3.88–+7.76

SD, standard deviation; D, diopter; SE, spherical equivalent.

**Table 2 jcm-11-04176-t002:** Distribution and invasion range of limbal dermoids in 41 patients.

Variable	Mean ± Standard Deviation	Range/%
Lesion location		
Temporal inferior	40	97.56%
Nasal inferior	1	2.44%
Lesions scope		
Central circular zone	25	60.98%
Paracentral annular zone	16	36.58%
Corneal limbus within 1 mm	1	2.44%
Number of hours invoved	3.24 ± 1.02	2–6
Complications, such as eyelid and ear		
Yes	16	39.02%
No	25	60.98%

**Table 3 jcm-11-04176-t003:** Corneal topographic parameters of 39 patients with limbal dermoids.

	K1 (D)	K2 (D)	Km (D)	Ast (D)
Mean ± SD	38.48 ± 2.12	43.29 ± 1.97	40.70 ± 1.48	4.80 ± 2.93
Range	32.3–42.2	40.3–47.7	36.0–43.6	0.4–11.2
Variance	4.49	3.88	2.19	8.59

**Table 4 jcm-11-04176-t004:** Comparison of anterior corneal surface parameters in 39 patients.

	CCmax (D)	CCmin (D)	CTmax (μm)	ASHmax (μm)
Within the range of limbal dermoids	43.33 ± 4.80	38.73 ± 3.72	685.08 ± 67.23	40.44 ± 25.13
On the diagonal	39.86 ± 2.52	38.80 ± 1.81	648.28 ± 40.87	18.05 ± 17.06
t	4.242	−0.097	3.347	9.093
*p*-value	0.000 **	0.923	0.002 **	0.000 **

** *p* < 0.01.

**Table 5 jcm-11-04176-t005:** Comparison of the refractive state between the CCZ and PCZ groups.

	Number of Hours Involved	Spherical (D)	Astigmatism (D)	Axis of Astigmatism (Degrees)	SE (D)
CCZ	3.60 ± 1.12	5.41 ± 3.32	4.74 ± 2.08	98.00 ± 56.99	3.04 ± 3.57
PCZ	2.69 ± 0.48	3.83 ± 2.48	3.33 ± 1.31	112.19 ± 60.77	2.16 ± 2.12
t	3.598	1.634	2.425	−0.758	0.986
*p-*value	0.001 **	0.110	0.020 *	0.453	0.330

* *p* < 0.05; ** *p* < 0.01.

**Table 6 jcm-11-04176-t006:** Comparison of the refractive state of different treatment durations.

	Treatment Duration (Months)	Spherical (D)	Astigmatism (D)	Axis of Astigmatism (Degrees)	SE (D)
>12 months/10	39.44 ± 24.77	4.93 ± 2.72	3.43 ± 1.24	109.50 ± 58.66	3.21 ± 2.70
≤12 months/31	1.55 ± 3.10	4.75 ± 3.24	4.44 ± 2.06	101.61 ± 58.84	2.53 ± 3.22
t	−4.825	−0.154	1.463	−0.369	−0.602
*p-*value	0.001 **	0.878	0.152	0.714	0.551

** *p* < 0.01.

**Table 7 jcm-11-04176-t007:** Factors related to visual recovery.

Variable	Mean ± SD	r	*p*-Value
Age (months)	65.17 ± 22.27	0.392	0.048 *
Surgical age (months)	26.79 ± 12.84	0.248	0.221
The extent of limbal dermoid invasion		0.110	0.594
CCZ	18		
PCZ	9		
The number of hours involved	3.26 ± 1.06	−0.155	0.450
Presence of amblyopia treatment intervention		−0.327	0.103
Yes	14		
No	13		
Time between amblyopia intervention and operation (months)	23.01 ± 16.80	−0.275	0.174
The duration of amblyopia treatment intervention (months)	15.38 ± 23.85	0.488	0.011 *
Spherical (D)	4.38 ± 3.15	−0.387	0.050
Astigmatism (D)	4.31 ± 2.16	−0.646	0.000 **
Axis of astigmatism (degrees)	105.37 ± 59.44	−0.154	0.451
SE (D)	2.23 ± 3.22	0.195	0.340

* *p* < 0.05; ** *p* < 0.01.

## Data Availability

The data presented in this study are available on request from the corresponding author.

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
