# Peer review of "Refractive Characteristics and Related Factors of Amblyopia after Lamellar Keratoscleroplasty in Children with Limbal Dermoids"

_jcm, 2022, doi:10.3390/jcm11144176_

Round 1
Reviewer 1 Report
This manuscript presents the refractive characteristics and related factors of amblyopia in pediatric patients with limbal dermoids undergoing lamellar keratoplasty. This is well written, and the research findings will be important for not only the general readers but the experts in this field.
The reviewer has nothing to mention to need to revise other than the limitations. This study lacks the comparison between preoperative and postoperative refractive data. The authors should mention that.
The reviewer hopes that this comments will be beneficial for publication to this journal.
Author Response
Response to Reviewer 1 Comments
Comments 1: The reviewer has nothing to mention to need to revise other than the limitations. This study lacks the comparison between preoperative and postoperative refractive data. The authors should mention that.
Response 1: We would like to thank the reviewer for evaluating our manuscript and for his/her comment. In this study, as the age of patients with limbal dermoids would affect our collection of refractive data before surgery, and in addition, some patients might have been lost in outpatient follow-up, we did not have complete preoperative and post-operative refractive data. We have discussed this issue in the revised manuscript and extended the limitation part as follows:
“Our study has some limitations. First, it was a cross-sectional study, and the relatively small sample size may have limited our ability to reach definitive conclusions. Second, the patients’ operation age was relatively young, and some patients might have been lost to follow-up. Therefore, there were no complete refractive data available both before and after surgery. Here, only the results of one outpatient examination of patients with limbal dermoids were selected, and the comparison of preoperative and postoperative diopters was not conducted. Third, long-term observations are required to expand the knowledge concerning the vision results of patients with limbal dermoids.” (Lines 311-318)
Reviewer 2 Report
The authors aimed to analyze the refractive characteristics and related factors of amblyopia in pediatric patients with limbal dermoid undergoing lamellar keratoscleroplasty and they found that high hyperopia and astigmatism are the dominant refractive errors in patients with limbal dermoid undergoing lamellar keratoscleroplasty.
The article is original and of interest to the scientific community of ophthalmologist and optometrist. Several comments should be solved prior to continue with the publication process.
Line 36 – 45 Please do not include general information abut corneal limbal dermoid, ophthalmologist that want to read this manuscript should know this information, be more precise.
Line 55 – How did you select the eye of the patient?
Please include the institutional review board approval
Include more details in the section Pentacam anterior segment analysis system
Why the corneal topography was only for 39 patients?, please explain this approach
Format and established all tables in the correct form and position in order to improve the visual appearance of the manuscript.
Extend the reference on the discussion
Include references only after 2010, when possible
Include references in indexed and impacted journals
Please provide sections in the discussion.
Rephrased lines from 226 to 249 to improve the comprehension of this part
The limitation section should extend with additional information
Include future research lines for probable future topics
Author Response
Response to Reviewer 2 Comments
Comments 1: Pg 1, Line 36–45: Please do not include general information abut corneal limbal dermoid, ophthalmologist that want to read this manuscript should know this information, be more precise.
Response 1: We would like to thank the reviewer for evaluating our manuscript and for his/her comment. Please note that we have removed the general information concerning corneal limbal dermoid and revised the corresponding part as follows:
“A corneal limbal dermoid is a congenital choristoma that exists at birth [1] and it can gradually increase with age [2]. Corneal limbal dermoid also has a relatively large impact on vision [3]. Surgery for limbal dermoid has been very successful by improving appearance and rarely causing rejection [4-6].” (Lines 35–38)
References:
- Sharma, A.; Sukhija, J.; Das, A.; Saroha, V.; Sukhi, S.; Mohan, K. Large pedunculated congenital corneal dermoid in association with eyelid coloboma. J. Pediatr. Ophthalmol. Strabismus. 2004, 41, 53–55. DOI:3928/0191-3913-20040101-12.
- Pirouzian, A. Management of pediatric corneal limbal dermoid. Clin. Ophthalmol. 2013, 7, 607–614. DOI:2147/OPTH.S38663.
- Zhong, J.; Deng, Y.; Zhang, P.; Li, S.; Huang, H.; Wang, B.; Zhang, H.; Peng, L.; Yang, R.; Xu, J.; et al. New grading system for limbal dermoid: A retrospective analysis of 261 cases over a 10-Year period. Cornea 2018, 37, 66–71. DOI:1097/ICO.0000000000001429.
- Jeong, J.; Song, Y.J.; Jung, S.I.; Kwon, J.W. New surgical approach for limbal dermoids in children: simple excision, corneal tattooing, and sutureless limboconjunctival autograft. Cornea. 2015, 34, 720– DOI:10.1097/ICO.0000000000000440.
- Scott, J.A.; Tan, D.T.H. Therapeutic lamellar keratoplasty for limbal d Ophthalmology 2001, 108, 1858–1867. DOI:10.1016/s0161-6420(01)00705-9.
- Shen, Y.D.; Chen, W.L.; Wang, I.J.; Hou, Y.C.; Hu, F.R. Full-thickness central corneal grafts in lamellar keratoscleroplasty to treat limbal dermoids. Ophthalmology 2005, 112, 1955–1962. DOI:1016/j.ophtha.2005.06.015.
Comments 2: Pg 2, Line 55: How did you select the eye of the patient?
Response 2: We would like to thank the reviewer for the question. All of the enrolled patients had a single eye corneal limbal dermoid. We examined the eye with corneal limbal dermoid. We have presented this information in the revised manuscript as follows:
“All of the enrolled patients had a single eye corneal limbal dermoid. We examined the eye with corneal limbal dermoid.” (Lines 54–55)
Comments 3: Please include the institutional review board approval.
Response 3: The study was approved by the hospital's ethics Committee on June 8, 2015. We have provided this information as follows:
“This study complied with the Declaration of Helsinki, and the parents of all patients were informed of the details of the study and provided their informed consent. The study was approved by the hospital's ethics Committee on June 8, 2015.” (Lines 56–58)
Comments 4: Include more details in the section Pentacam anterior segment analysis system.
Response 4: We have provided more information as follows:
“The Pentacam is an automatic non-contact device that utilizes a rotating Scheimpflug camera and monochromatic slit illumination to provide video-based automated anterior segment measurements. The camera rotates around the eye while taking images, and with the help of an internal software, it generates a three-dimensional model of the anterior segment. The anterior segment model consists of high-resolution imaging that allows for assessment of the entire surface structure of the cornea as well as the measurement of the anterior chamber depth. Pentacam performed the measurement automatically when correct alignment with the corneal apex and focus was achieved. The noninvasive measurement process of the Pentacam requires 2 s and collects 25–50 single captures that consist of 13,800 true elevation points while rotating around the optical axis of the eye. This results in a Pentacam built three-dimensional model of the entire anterior eye segment.
Records of patients examined with the Scheimpflug principle-based Pentacam corneal tomographer were reviewed for topographic parameters. Parameters studied included keratometry readings, topographic astigmatism, corneal eccentricity in the central 6 mm, average radius of curvature between the 6-mm and 9-mm zone center, and minimum sagittal curvature for anterior and posterior cornea surfaces.” (Lines 86-102)
Comments 5: Why the corneal topography was only for 39 patients?, please explain this approach.
Response 5: We would like to thank the reviewer for the question. Two patients were too young to complete the corneal topography examination and, therefore, we performed corneal topography examination in 39 patients. We have provided this information in the revised manuscript as follows:
“Two patients were too young to complete the corneal topography examination and, therefore, we completed the corneal topography examination in 39 patients.” (Lines 83–85)
Comments 6: Format and established all tables in the correct form and position in order to improve the visual appearance of the manuscript.
Response 6: We would like to thank the reviewer for the beneficial comment. We have made the appropriate changes, as per the reviewer’s suggestion.
Comments 7: Extend the reference on the discussion.
Response 7: We would like to thank the reviewer for the constructive comment. We have reviewed the available works that focused on corneal limbal dermoids, which were reported in the last 10 years, and studied the content of the articles. Some modifications and supplements were made in the Discussion section.
Please note that we have cited the following studies in the revised manuscript:
References:
- Zhong, J.; Deng, Y.; Zhang, P.; Li, S.; Huang, H.; Wang, B.; Zhang, H.; Peng, L.; Yang, R.; Xu, J.; et al. New grading system for limbal dermoid: A retrospective analysis of 261 cases over a 10-Year period. Cornea 2018, 37, 66– DOI:10.1097/ICO.0000000000001429.
- Jeong, J.; Song, Y.J.; Jung, S.I.; Kwon, J.W. New surgical approach for limbal dermoids in children: simple excision, corneal tattooing, and sutureless limboconjunctival autograft. Cornea. 2015, 34, 720– DOI:10.1097/ICO.0000000000000440.
- del Rocio Arce Gonzalez, M.; Navas, A.; Haber, A.; Ramirez-Luquin, T.; Graue-Hernandez, E.O. Ocular dermoids: 116 consecutive cases. Eye Contact Lens, 2013, 39, 188– DOI:10.1097/ICL.0b013e31824828ee.
- Zhong, J.; Wang, W.; Li, J.; Wang, Y.; Hu, X.; Feng, L.; Ye, Q.; Luo, Y.; Zhu, Z.; Li, J.; et al. Effects of perceptual learning on deprivation amblyopia in children with limbal dermoid: a randomized controlled trial. J. Clin. Med. 2022, 11, 1879. DOI: 3390/jcm11071879.
Comments 8: Include references only after 2010, when possible.
Response 8: We would like to thank the reviewer for the insightful comment. We tried our best to select references related to the topic of our manuscript that were reported within the last 10 years, as they are more novel and more consistent with the current clinical practice in terms of content, topic, or research direction. However, some of the older articles are very classic and have a positive effect on the writing of the article after reading.
Comments 9: Include references in indexed and impacted journals.
Response 9: We would like to thank the reviewer for the constructive comment. Most of the references cited by us are published in influential journals and can be obtained through a search in the PubMed database.
Comments 10: Please provide sections in the discussion.
Response 10: We would like to thank the reviewer for the suggestion. Please note that we have added a subsection (4.1. Limitations) in the Discussion section.
Comments 11: Rephrased lines from 226 to 249 to improve the comprehension of this part.
Response 11: We would like to thank the reviewer for the suggestion. Please note that we have revised this part as follows:
“The article also mentioned that preoperative treatment of amblyopia can improve postoperative visual acuity, and long-term observation and follow-up examinations are required for preoperative and postoperative amblyopia treatment.
During the postoperative follow-up period, the patients were examined by Pentacam corneal topography and anterior segment photography. Their visual acuity, spherical lens refraction, and astigmatism were recorded. The corneal topographic characteristics and distribution characteristics of the refractive state were summarized, and the relationship between the extent of limbal dermoid invasion, corneal morphology, and refractive state was analyzed. Furthermore, the relationship between the size of the limbal dermoid invasion and amblyopia formation as well as the effect of clinical intervention at different times after amblyopia formation was also examined. We found that most patients experienced hyperopia, which may be partly attributed to the normal visual development of young children. The clinical presentation of a limbal dermoid is a hemispherical or fusiform bulging mass at the margin of the corneal limbus. The extent of corneal invasion directly affects the morphology of the cornea and abnormal corneal morphology leads to abnormal diopters. Lamellar keratoscleroplasty does not completely correct the abnormal corneal morphology and refraction. Therefore, high astigmatism is mostly caused by corneal morphological changes because of the presence of limbal dermoids in patients. Hussein et al. [12] concluded that initial low vision, age >6 years, obvious astigmatism, and poor treatment compliance are the factors that can lead to the failure of anisometropic amblyopia treatment. However, in a paper published by the ophthalmic surface team of the Fudan University Eye, Ear, Nose, and Throat Hospital [13], it was found that residual and surgically induced equivalent spherical diameter were both reduced in long-term amblyopia treatment, but the correction effect on postoperative astigmatism was disappointing.” (Lines 246–271)
Comments 12: The limitation section should extend with additional information.
Response 12: We would like to thank the reviewer for the suggestion. Please note that we have extended the limitations as follows:
“Our study has some limitations. First, it was a cross-sectional study, and the relatively small sample size may have limited our ability to reach definitive conclusions. Second, the patients’ operation age was relatively young, and some patients might have been lost to follow-up. Therefore, there were no complete refractive data available both before and after surgery. Here, only the results of one outpatient examination of patients with limbal dermoids were selected, and the comparison of preoperative and postoperative diopters was not conducted. Third, long-term observations are required to expand the knowledge concerning the vision results of patients with limbal dermoids.” (Lines 311-318)
Comments 13: Include future research lines for probable future topics.
Response 13: We would like to thank the reviewer for the insightful suggestion. We have added relevant information in the article as follows:
“As surgery cannot change the high refractive abnormality, especially the astigmatism, we should seek ways to reduce the astigmatism of patients, including adjusting the tightness of sutures and removal of intermittent stitches as well as postoperative optical correction methods (i.e., wearing RGP lenses) and active amblyopia treatment methods (i.e., perceptual learning) [17].” (Lines 304-308)
Reference:
- Zhong, J.; Wang, W.; Li, J.; Wang, Y.; Hu, X.; Feng, L.; Ye, Q.; Luo, Y.; Zhu, Z.; Li, J.; et al. Effects of perceptual learning on deprivation amblyopia in children with limbal dermoid: a randomized controlled trial. J. Clin. Med. 2022, 11, 1879. DOI: 3390/jcm11071879.
